# Two-way text message interventions and healthcare outcomes in Africa: Systematic review of randomized trials with meta-analyses on appointment attendance and medicine adherence

**Emilie S. Ødegård[1]☉, Lena S. Langbråten[1]☉, Andreas Lundh ⓘ[2,3,4], Ditte S. Linde ⓘ[1,5] ***

**1** Department of Clinical Research, University of Southern Denmark, Odense, Denmark, **2** Odense Patient Data Explorative Network (OPEN), Odense University Hospital, Odense, Denmark, **3** Centre for Evidence-Based Medicine Odense (CEBMO) and Cochrane Denmark, Department of Clinical Research, University of Southern Denmark, Odense, Denmark, **4** Department of Infectious Diseases, Hvidovre Hospital, Hvidovre, Denmark, **5** Department of Obstetrics and Gynaecology, Odense University Hospital, Odense, Denmark

☉ These authors contributed equally to this work.
* dsondergaard@health.sdu.dk

**Citation:** Ødegård ES, Langbråten LS, Lundh A, Linde DS (2022) Two-way text message interventions and healthcare outcomes in Africa: Systematic review of randomized trials with meta-analyses on appointment attendance and medicine adherence. PLoS ONE 17(4): e0266717. https://doi.org/10.1371/journal.pone.0266717

**Data Availability Statement:** All relevant data are within the paper and its Supporting information files.

## Abstract

### Background

The growing access to mobile phones in Africa has led to an increase in mobile health interventions, including an increasing number of two-way text message interventions. However, their effect on healthcare outcomes in an African context is uncertain. This systematic review aims to landscape randomized trials involving two-way text message interventions and estimate their effect on healthcare outcomes.

### Methods

We searched Medline, Embase, Cochrane Central Register of Controlled Trials, The Global Health Library (up to 12 August 2021) and trial registries (up to 24 April 2020). Published and unpublished trials conducted in Africa comparing two-way text message interventions with standard care and/or one-way text message interventions were included. Trials that reported dichotomous effect estimates on healthcare appointment attendance and/or medicine adherence were assessed for risk of bias and included in meta-analyses. Results of other outcomes were reported descriptively.

### Results

We included 31 trials (28,563 participants) all set in Sub-Saharan Africa with a wide range of clinical conditions. Overall, ten different trials were included in the primary meta-analyses, and two of these had data on both medicine adherence and appointment attendance. An additional two trials were included in sensitivity analyses. Of the 12 included trials, three were judged as overall low risk of bias and nine as overall high risk of bias trials. Two-way

**Funding:** The authors received no specific funding for this work.

**Competing interests:** The authors have declared that no competing interests exist.

**Abbreviations:** CI, Confidence Interval; cMMs, Community mentor mother; eHealth, Electronic health; Hb1Ac, Haemoglobin A1c; HIV, Human Immunodeficiency virus; HR, Hazard ratio; I, Imaginary number; ISRCTN, International Standard Randomized Controlled Trial Number Registry; LMICs, Low- and middle-income countries; MAST, Model for assessment of telemedical applications; mHealth, Mobile health; OR, Odds Ratio; PACTR, Pan African Clinical Trials Registry; PRISMA, Preferred Reporting Items for Systematic Reviews and Meta-Analyses; RCT, Randomized controlled trial; RD, Risk difference; RR, Risk Ratio; SMS, Short message service; UN, United Nations; WHO, World Health Organization.

text messages did not improve appointment attendance, RR: 1.03; 95% CI: 0.95–1.12, $I^2 = 53\%$ (5 trials, 4374 participants) but improved medicine adherence compared to standard care, RR: 1.14, 95% CI: 1.07–1.21, $I^2 = 8\%$ (6 trials, 2783 participants).

## Conclusion

Two-way text messages seemingly improve medicine adherence but has an uncertain effect on appointment attendance compared to standard care.

## Systematic review registration

PROSPERO CRD42020175810.

## Introduction

For the past 30 years, information and communication technologies have transformed the world [1]. In recent years mobile phone access has expanded immensely across Africa and other low- and middle-income countries (LMICs). According to the World Health Organization (WHO), more people have access to a mobile phone than to clean running water in sub-Saharan Africa [2]. In 2017, approximately 8 out of 10 people in sub-Saharan Africa owned a mobile phone with an increasing amount of these being smartphones [3]. The growing access to mobile phones has led to new possibilities in relation to health, i.e., mobile health (mHealth), which comprises a wide range of communication channels such as mobile phone surveys, mobile phone calls or Short Message Service (SMS)/text messages between clients and health practitioners [2, 4]. One-way text messages are messages, which the receiver cannot respond to, while two-way messages are a form of interactive communication where the text message receiver can respond to the message or in other ways interact with the sender, e.g., by participating in a text message quiz, responding to the text message, calling the sender or receiving a phone call from the sender [5].

A systematic review and meta-analysis from 2015 –which included trials conducted in various settings–suggested that interactive communication between a client and a healthcare provider may lead to greater support, motivation, and safety for the patient. The meta-analysis found a higher intervention effect of two-way messages compared to one-way text messages on medicine adherence [6]. Further, a series of Cochrane reviews published between 2012 and 2017 assessed the effect of various text message interventions on different health issues [7–12], but the reviews did not differentiate between one-way and two-way text message interventions in their analyses, hence, the effect of one-way versus two-way text messages interventions was unclear. Additionally, most trials included in these reviews were conducted in high-income countries and only two reviews included trials from Africa [10, 11]. However, the geographical setting may be an important factor to account for in relation to the effect of text message interventions as digital literacy, network infrastructure, and cultural/social acceptance of mHealth interventions may differ across regions [2]. A systematic review and meta-analysis from 2019 [5] landscaped randomized controlled trials (RCTs) of one-way text message interventions in Africa and reported an effect on healthcare appointment attendance, OR: 2.03; 95% CI: 1.40–2.95 whilst the effect was uncertain on medicine adherence, RR 1.10; 95% CI: 0.98–1.23. Research shows that linkage to care is a challenge in many resource-limited settings [13], and it is plausible that two-way text messages–which allow for interactive communication with

health care providers–may be a more effective health care tool than one-way messages. Appointment attendance and medicine adherence are outcomes frequently used to measure linkage to care across clinical settings, yet the effect of two-way messages on these outcomes in African context is unclear. Therefore, this systematic review aims to landscape randomized trials in Africa involving two-way text message interventions compared to standard care or one-way text messages and analyze their effect on appointment attendance and medicine adherence. By conducting a meta-analysis on the same outcomes as the previous systematic review of one-way messages in Africa [5], this review will provide an even greater understanding of how various types of text message interventions work in an African context. Further, in line with the previous review, we also describe other health outcomes to provide a full overview of two-way message trials set in Africa.

## Materials and methods

### Protocol registration

This systematic review is based on a protocol reported according to the Preferred Reporting Items for Systematic Reviews and Meta-Analyses Protocol (PRISMA-P) guidelines [14] (S1 File) and registered prior to review conduct (PROSPERO CRD42020175810).

### Eligibility criteria

We included randomized trials of male and female healthcare clients, partners of or guardians for healthcare clients, e.g., parents for child patients. Trials targeting health personnel and healthcare providers were excluded. Trials comparing two-way text message interventions with standard care and/or one-way text messages were included. We defined two-way text messages as a text message the receiver could react or respond to, with the use of a mobile phone, in any way. If trials were multi-arm, e.g., included one-way and two-way interventions as well as standard care, these were included and data for each arm were analyzed separately. If co-interventions (e.g., written material) were received by participants in both the intervention and control arms, this was considered to be part of standard care, and such trials were included. We included published and unpublished randomized trials in any language conducted in Africa, including pilot and cluster trials. International multicenter trials that had African sites were included if separate results for the African sites were reported or could be accessed upon request. Trials reporting all healthcare outcomes were included if they met all other eligibility criteria.

### Information sources and search strategy

We searched Medline, Embase, Cochrane Central Register of Controlled Trials and The Global Health Library until 12 August 2021. The search strategy was adapted from a strategy used in our previous systematic review of one-way text message interventions in Africa [5]. We tailored the search strategy for each database with the support of an information specialist. The search string included search terms such as "randomized controlled trial", "Africa", "SMS", "mobile phone", "two-way text messages" and "two-way SMS" (S2 File). ClinicalTrials.gov, Pan African Clinical Trial Registry (PACTR) and The International Standard Randomised Controlled Trial Number registry (ISRCTN) were searched up to 24 April 2020 for additional trials, including ongoing and unpublished trials. Reference lists of relevant systematic reviews and of the included trials were searched for additional trials. Finally, we searched the UN and World Bank reports for additional trials (up to 18 March 2020). In August 2021 the status of ongoing trials was checked for published results.

Unique records were uploaded to Covidence (www.covidence.org). Two authors (ESØ, LSL) independently screened titles and abstracts of all retrieved records. In the title-abstract screening, trials were only excluded if the title and abstract obviously stated that interventions consisted of one-way messages as according to our definition. Trials selected for full text evaluation were independently assessed by two authors (ESØ, LSL) for inclusion. Two authors (ESØ, LSL) extracted data, and each author extracted data from half of the included articles into a standardized Excel data extraction sheet and verified each other´s data extraction for trial outcomes. Disagreements in relation to trial inclusion and data extraction were resolved through discussion, and if consensus could not be reached an arbiter (DSL) made the final decision. Extracted data included: first author, title, publication year, name of journal/registry, objective, inclusion/ and exclusion criteria, randomization method, trial period, sample size, description of experimental and control interventions, primary target group, pre-test and theoretical foundation of intervention, length of follow-up, trial outcomes, results for primary trial outcome and results for appointment attendance and medicine adherence. Two authors (ESØ, LSL) contacted trial authors for missing data, including data from unpublished trials.

## Risk of bias assessment

Two authors (ESØ, LSL) independently assessed all trials that were eligible for meta-analysis (trials reporting the outcomes "appointment attendance" and/or "medicine adherence" for risk of bias using the Cochrane Risk of Bias Tool [15]. If consensus could not be reached, and arbiter (DSL) made the final decision. The items assessed were random sequence generation and allocation concealment (selection bias), blinding of participants and personnel (performance bias), blinding of outcome assessment (detection bias), incomplete outcome data (attrition bias), selective reporting (reporting bias) and other biases. As part of the other biases assessment "The Beall's list of potential predatory publishers" and the "List of Predatory journals" [16, 17] were searched, and trials that were published in a potential predatory journal were judged as high risk of other biases. For cluster randomized trials, risk of bias related to recruitment and baseline imbalance were also assessed [15]. The items were judged to have low, high, or unclear risk of bias. However, attrition bias was not assessed for trials that only reported "appointment attendance" as an outcome as incomplete data is an integral part of this outcome, i.e., non-attendance is equivalent to loss to follow-up. If trials were judged to have low risk of selection, detection and reporting bias, they were judged to have overall low risk of bias. If not, they were judged to have overall high risk of bias.

## Data analysis

Two-way text message interventions were expected to be used in various types of populations and settings and for multiple types of outcomes. Therefore, we conducted an overall descriptive analysis of all trials and restricted meta-analyses to the outcomes "appointment attendance" and "medicine adherence", which we regarded to be uniform. For our descriptive analysis, we reported unadjusted trial results for the primary outcome, unless the trial only reported adjusted results. If no quantitative estimates were reported, but binary data were available, risk ratios (RRs) and 95% confidence intervals were calculated.

Meta-analyses were done using Review Manager 5.4.1 [18]. Due to the anticipated clinical and methodological heterogeneity, we planned to calculate pooled RRs and estimate 95% confidence intervals (CIs) using a random-effects model with the Mantel-Haenszel method for dichotomous data, for both appointment attendance and medicine adherence. However, two trials assessing appointment attendance [19, 20], and one trial assessing medicine adherence [21], were randomized at cluster level, and we therefore used the inverse variance method for

our analyses. We only included trials with results adjusted for clustering in our primary analysis. Heterogeneity was assessed using $I^2$ [15]. We conducted separate analyses comparing two-way text messages to one-way text messages. Subgroup analyses were performed comparing overall low risk of bias trials with high risk of bias trials, clinical areas and types of interventions, e.g. if the receiver could interact with a health care professional (HCP) through a phone call or text message. Further, sensitivity analyses were performed using the fixed-effect models and excluding the cluster randomized trials [19–21]. In addition, one of the three cluster trials did not adjust for clustering [21] and was excluded from our primary analysis but included in an additional sensitivity analysis. We intended to exclude trials published in potential predatory journals in a sensitivity analysis, but all trials were published in legitimate journals. Trials where we were unable to extract the required dichotomous event and participant data or trials that only reported continuous outcomes were excluded from the meta-analysis and analyzed descriptively. If adherence was measured in multiple ways in a trial, the most objective outcome measure was chosen for the meta-analysis, e.g. pill box openings were prioritized over self-reported adherence. However, dichotomous self-reported outcomes were used if it was the only outcome measure apart from a continuous outcome.

## Results

### Study selection

We identified 2624 records in our database search. After removal of duplicates there were 2150 unique records of which 2068 were excluded based on title- or abstracts (Fig 1). Eighty-two records were eligible for full text evaluation and a total of 29 trials were eligible to be included in the review [19–47], however, one multicenter trial was excluded because of no individual data on the African settings [42]. Further, searching other sources led to the inclusion of 3 additional trials [48–50]. In total, we included 31 trials, of which 20 were published [19–35, 43–45] and 11 were unpublished; six of the unpublished trials were finished [36–38, 41, 48, 49] and 5 had an unknown status [39, 40, 46, 47, 50].

### Study characteristics

Trials were published between 2010 and 2021 and included a total of 16,924 participants (median: 846 participants per trial) [19–35, 43–45] (Table 1). For the eleven unpublished trials, ten trials planned to enroll a total of 11,639 participants (median: 1164 participants per trial) [36–41, 46–50] (Table 1), and one remaining trial did not report the planned number of participants to be enrolled [49]. Eighteen corresponding authors were contacted for missing data including data from unpublished trials, and ten replied [28, 33–35, 37, 40–42, 48, 50]. Relevant queries were answered, but none of the respondents provided us with missing data or unpublished results.

The mean age among trial participants ranged from 17 to 54 years. The primary target group for the text-message interventions were guardians of infant patients [19, 20, 24, 25, 36, 37, 39, 40, 46, 47] where the mother was the receiver of the text message intervention, whilst the remaining trials solely targeted men [22, 28] or women [32, 33, 35, 44, 46, 48–50] or both men and women [23, 26, 27, 29–31, 34, 38, 41, 43, 45]. All trials were set in Sub-Saharan Africa with vast majority being set in Kenya (n = 17) followed by South Africa (n = 4) (Fig 2). Participants were enrolled from a wide range of settings, e.g., local health facilities, public hospitals, HIV clinics, schools, and privately-owned pharmacies. The clinical areas in trials were HIV (n = 20), reproductive health (n = 4), antenatal health (n = 4), non-communicable diseases (n = 2), and malaria (n = 1). Ten trials reported that the text message content was developed based on health behavioral theories [19, 24, 28, 30, 32, 34, 35, 37–39], eight reported that the

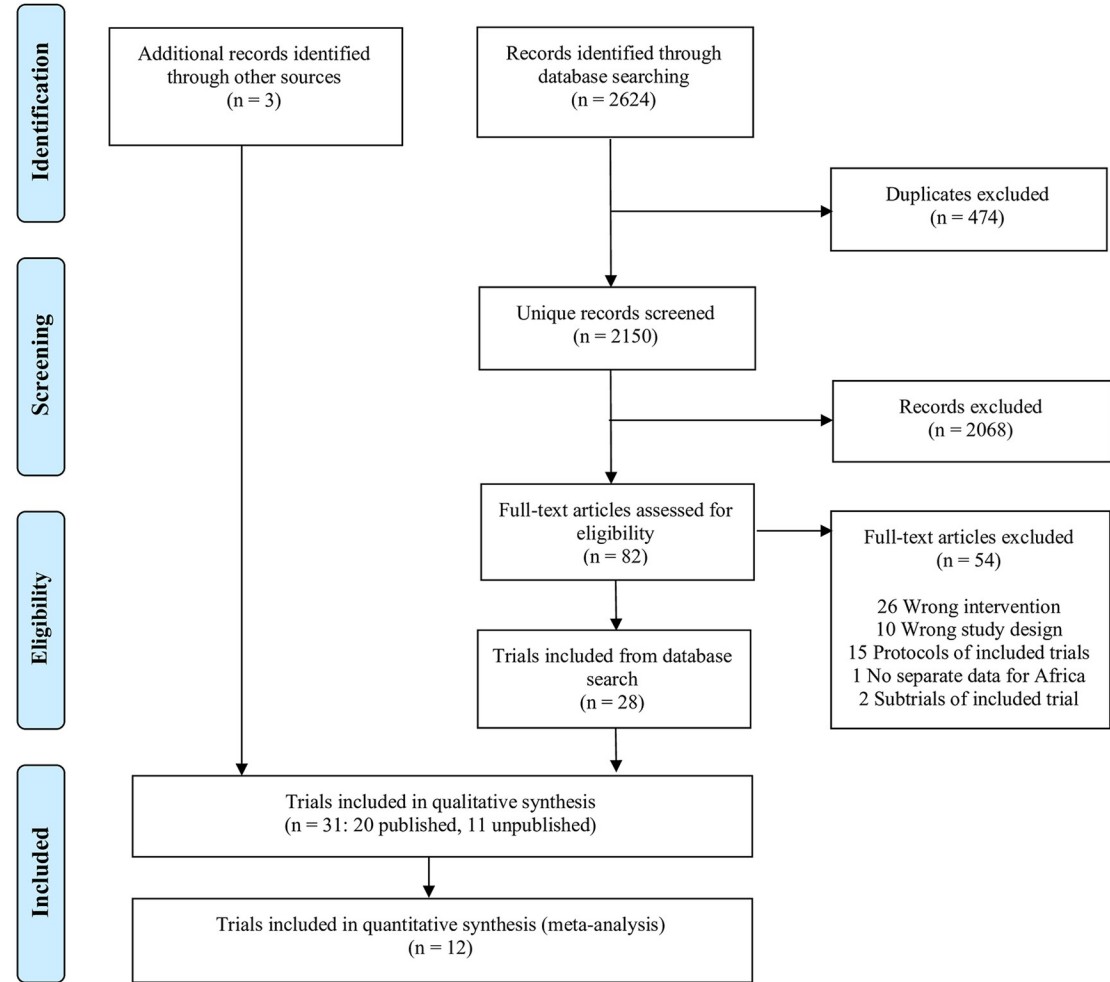

**Fig 1. Flow diagram.** *From*: Moher D, Liberati A, Tetzlaff J, Altman DG, The PRISMA Group (2009). *Preferred Reporting Items for Systematic Reviews and Meta-Analyses: The PRISMA Statement. PLoS Med 6(7): e1000097. doi:10.1371/journal.pmed1000097. **For more information, visit** www.prisma-statement.org.

content had been pre-tested or developed in consultation with experts, clinical staff and/or potential participants [21, 23, 25, 26, 31, 39, 44, 46] and thirteen trials did not report either [20, 22, 27, 29, 33, 36, 41, 43, 45, 47–50]. The content of the two-way text messages varied from supportive, educational, and motivational messages to reminders and quizzes with a two-way component of either replying to the message via text message or phone call or by requesting phone calls or additional information from the sender (Table 1).

When looking across all trials, ten trials included appointment attendance as an outcome [19, 20, 24–26, 28, 31, 39, 41, 44] and sixteen trials included medicine adherence as an outcome [21, 22, 25, 27, 29–31, 36, 37, 39, 41, 43–45, 48, 49]. Medicine adherence was measured in multiple ways, including self-reported adherence, pill box openings and clinical outcomes affected by adherence such as change in mean systolic blood pressure, proportion of patients with

**Table 1. Two-way text message trials set in Africa.**

| Publication year, author | Country | Clinical area | Trial size (n) | Female participants | Age (mean) | Follow-up length (weeks) | Primary outcome | Type of two-way intervention | Effect of two-way intervention on primary outcome compared to control |
|---|---|---|---|---|---|---|---|---|---|
| | | | | | | | | Type of one-way intervention | Effect of two-way intervention on primary outcome compared to one-way intervention |
| *Finished trials, published* | | | | | | | | | |
| 2021, Abiodun [43] | Nigeria | HIV | 212 | 48% | 16.6 | 20 | Medicine adherence | Appointment and medicine adherence reminder + text message response option | RR: 1.2 [95% CI: 0.9 to 1.6] (VAS score ≥95% adherence) |
| | | | | | | | | Appointment reminders (control group) | - |
| 2021, Kinuthia [44] | Kenya | HIV | 824 | 100% | 27 | 24 | (1)Appointment attendance (2) Medicine adherence | Reminders + Educational + text message response option | aRR$_{Appointment\ attendance}$: 1.0 [95% CI: 1.0 to 1.0] ¶ |
| | | | | | | | | | aRR$_{Medicine\ adherence}$: 0.8 [95% CI: 0.5 to 1.2]¶ |
| | | | | | | | | Reminders + Educational | aRR$_{Appointment\ attendance}$: 1.0 [95% CI: 1.0 to 1.0] ¶ |
| | | | | | | | | | aRR$_{Medicine\ adherence}$: 0.8 [95% CI: 0.5 to 1.2]¶ |
| 2021, Sumari-de Boer [45] | Tanzania | HIV | 245 | 71% | 41,2 | 48 | Medicine adherence | Reminders + text message response option | RR††: 1.1 [95% CI: 0.8 to 1.4] |
| 2020, Ampt [35] | Kenya | Unintended pregnancies among sex workers | 882 | 100% | 25.4 | 52 | Incidence of unintended pregnancy | Educational and motivational with text message response option and on-demand additional messages | HR*: 1.0 [95% CI: 0.7 to 1.4] |
| 2019, Feldacker [22] | Zimbabwe | HIV-related circumcision | 722 | 0% | 24 | 2 | Adverse event rate | Supportive with text message response option | Rate$_{intervention}$: 6/320; Rate$_{control}$: 3/359 |
| | | | | | | | | | RD†: 1.04% (p = 0.32) |
| 2019, Harrington [32] | Kenya | Post-partum contraception | 260 | 100%ᵞ | 22.8 | 38 | Post-partum contraceptive use | Educational with text message response option | RR‡: 1.2 [95% CI: 1.0 to 1.5] |
| 2019, Odeny [19] | Kenya | HIV and post-partum care | 2515 (20 clusters) | 100% *HIV-positive mothers* | 27 | 8 | (1) Infant HIV testing; (2) post-partum appointment attendance | Supportive + educational + reminders + text message/phone call option | RR$_{HIV-testing}$: 1.1 [95% CI: 1.0 to 1.1] |
| | | | | | | | | | RR$_{post-partum\ attendance}$: 1.2 [95% CI: 1.0 to 1.3] |
| 2018, Unger [25] | Kenya | Antenatal care | 298 | 100% *Mothers* | 23 | 24 | Delivery at health facility | Individual tailored educational + motivational + quiz | RR: 1.0 [95% CI: 1.0 to 1.0] |
| | | | | | | | | Individual tailored educational + motivational | - |
| 2018, Van der Kop [26] | Kenya | HIV | 700 | 60% | 33.7 | 55 | Appointment attendance | Supportive with text message response option | RR: 1.0 [95% CI: 0.9 to 1.1] |
| 2017, Linnemayr [27] | Uganda | HIV | 332 | 61% | 18.3 | 48 | Medicine adherence | Supportive with text message response option | Mean adherence‡ $_{2-way\ intervention\ vs.\ control}$ = 0.61 vs. 0.67, (p = 0.15) |
| | | | | | | | | Supportive | - |
| 2017, Rokicki [33] | Ghana | Sexual reproductive health | 756 (38 clusters) | 100% | 17.7 | 15 | Reproductive health knowledge | Educational + quiz where correct answer is sent after responding to the quiz | 24% higher score than control group [95% CI: 19 to 28] |
| | | | | | | | | Educational | 13% higher score than one-way intervention [95% CI: 8 to 18] |
| 2017, Van Olmen [34] | Congo Cambodia Philippines | Diabetes type 1 and 2 | 1471 Congo: 506 | 66% | 60 | 104 | Diabetes control (Hb1Ac < 7%) | Educational + supportive + text message/phone call option | RD$^§$ $_{diabetes\ control,\ all\ countries}$ = 2,7% (p = 0.4) |
| | | | | | | | | | RD$^§$ $_{diabetes\ control,\ Congo}$ = 7,5% |
| 2016, Bobrow [23] | South Africa | Hypertension | 1372 | 72% | 54.3 | 52 | Change in mean systolic blood pressure | Educational + motivational + reminders + phone call option | RD: -1.6 mmHg [95% CI: −3.7 to—0.7] |
| | | | | | | | | Educational + motivational + reminders | - |

*(Continued)*

**Table 1.** (Continued)

| Publication year, author | Country | Clinical area | Trial size (n) | Female participants | Age (mean) | Follow-up length (weeks) | Primary outcome | Type of two-way intervention | Effect of two-way intervention on primary outcome compared to control |
| --- | --- | --- | --- | --- | --- | --- | --- | --- | --- |
| | | | | | | | | Type of one-way intervention | Effect of two-way intervention on primary outcome compared to one-way intervention |
| 2016, Kassaye [21] | Kenya | HIV | 550 (26 clusters) | 100% Mothers | 25.6 | 6 | Mothers medicine adherence | Educational + motivational + supportive + reminders + text message/phone call option | aRR¶: 1.0 [95% CI: 0.9 to 1.2] |
| 2016, Leiby [28] | Zambia | HIV-related circumcision | 1652 | 0% | 23 [median] | 26 | Appointment attendance | Educational + text message option [Conventional two-way intervention] | OR ‖: 1.1 [95% CI: 0.8 to 1.7] |
| | | | | | | | | Educational + text message option [Tailored two-way intervention] | - |
| 2014, Lund [20] | Tanzania | Antenatal care | 2550 (24 clusters) | 100% Mothers | 27 [median] | 6 (post-partum) | Appointment attendance | Educational + reminders + phone call voucher | OR$_{appointment\ attendance}$:1.5 [95% CI: 0.8 to 3.0] |
| 2014, Modrek [29] | Nigeria | Malaria | 457 | 46% | 39 | 4 days | Medicine adherence | Reminder + phone call option | OR: 2.1 [95% CI: 1.4 to 3.0] |
| 2014, Odeny [24] | Kenya | HIV | 388 | 100% Mothers | 29 [median] | 8 | (1) Infant HIV testing; (2) post-partum appointment attendance | Supportive + educational + reminders + phone call option | RR$_{HIV-testing}$: 1.1 [95% CI: 1.0 to 1.2] |
| | | | | | | | | | RR$_{post-partum\ attendance}$: 1.7 [95% CI: 1.0 to 2.7] |
| 2012, Mbuagbaw [30] | Cameroon | HIV | 200 | 74% | 40.2 | 26 | Medicine adherence | Motivational + reminder + phone call option | RR: 1.1 [95% CI: 0.9 to 1.3] |
| 2010, Lester [31] | Kenya | HIV | 538 | 65% | 36.7 | 52 | Medicine adherence | Supportive with text message response option | RR: 1.2 [95% CI: 1.0–1.4] |
| *Finished trials, unpublished* | | | | | | | | | |
| 2018, Gonsalves [38] | Kenya Peru | Sexual and reproductive health | 1395 | - | - | 8 | Sexual and reproductive health knowledge | Educational + quiz + access to previous educational domains [Two-way intervention A] | - |
| | | | | | | | | Educational + quiz [Two-way intervention B] | - |
| 2018, Odeny [37] | Kenya | HIV | 1338 | 100% Mothers | - | 12 | Medicine adherence | Educational + supportive + call back option | - |
| | | | | | | | | Community mentor mother support | - |
| | | | | | | | | Educational + supportive + call back option + community mentor mother support | - |
| 2016, Lippman** [41] | South Africa | HIV | 752 (18 clusters) | 39% | 41,5 | 12 | Medicine adherence | Educational + reminders + phone call option | OR: 1 [95% CI: 0.6 to1.5] |
| | | | | | | | | Peer navigators | OR: 1.5 [0.9 to 2.5] |
| 2016, Awiti [36] | Kenya | HIV | 600 (planned) | 100% Mothers | - | 30 | Medicine adherence | Supportive with text message response option | - |
| 2015, NCT02627365** [48] | Kenya | HIV | 119 | 100% | - | 6 | Medicine adherence | Educational + motivational + response option | - |
| 2010, NCT01157442 [49] | Kenya | HIV | - | 100% | - | 12 | Medicine adherence | Reminders + text message response option | - |
| *Trials with unknown status, unpublished* | | | | | | | | | |
| 2020, Unger [46] | Kenya | Antenatal care | 5000 (planned) | 100% | - | 28–36 weeks gestation to 6 weeks post partum | Neonatal mortality | Educational + text message response option | - |

*(Continued)*

**Table 1.** (Continued)

| Publication year, author | Country | Clinical area | Trial size (n) | Female participants | Age (mean) | Follow-up length (weeks) | Primary outcome | Type of two-way intervention | Effect of two-way intervention on primary outcome compared to control |
|---|---|---|---|---|---|---|---|---|---|
| | | | | | | | | Type of one-way intervention | Effect of two-way intervention on primary outcome compared to one-way intervention |
| 2019, NCT04038060 [50] | South Africa | HIV | 350 (planned) | 100% | - | 12 | Medicine adherence | Supportive text message + counselling or drug level feedback | - |
| | | | | | | | | Supportive WhatsApp group + counselling or drug level feedback | - |
| 2019, Tickell [47] | Kenya | Malnutrition | 1200 (planned) | - | - | 6 | Time to diagnosis of acute malnutrition | Reminders + text message response option | - |
| 2017, Drake [39] | Kenya | HIV | 825 (planned) | 100% Mothers | - | 24 | Medicine adherence | Educational + motivational reminders + quiz | - |
| | | | | | | | | Educational + motivational reminders | - |
| 2017, Zunza [40] | South Africa | HIV | 60 (planned) | 100% Mothers | - | 24 | Breastfeeding adherence | Motivational + reminders + phone call option + motivational interviews | - |

*HR: Hazard ratio;

†RD: Risk difference;

§Only p-value reported/no p-value reported;

¶Only adjusted results reported;

‖OR; Odds Ratio

**Reported results from Clinicaltrials.gov/ISRCTN

††RR calculated based on 90% pharmacy refill (median), stated in table 4 in [45]

HbA1c < 7%, or suppressed HIV viral load [21, 25, 28, 29, 32, 34, 35, 38, 40, 42, 44]. Other trials had reproductive health knowledge [31, 39], adverse events [20], neonatal mortality [46], malnutrition [47], or unintended pregnancy as outcomes [33] (Table 1).

## Descriptive analyses of trials not included in meta-analysis

Eight trials reported other outcomes or reported results on medicine adherence and/or appointment attendance in a format that did not allow us to include the data in meta-analyses. Therefore, these outcomes were only analyzed descriptively.

Overall, results varied among the trials. Four trials reported outcomes related to reproductive health; One trial found that two-way text messages increased postpartum contraceptive use compared to standard care (RR: 1.2; 95% CI: 1.0–1.5) [32]. One three-armed cluster randomized trial investigated the effect on reproductive knowledge among adolescent girls in Ghana. The two-way text message group had 24% (95% CI: 19%-28%) higher questionnaire scores than the control group and 13% (95% CI: 8–18%) higher than the one-way text message group [33]. Another trial from Kenya assessed the effect of one-way and two-way text messages on facility delivery, exclusive breastfeeding, and contraceptive use. Compared to standard care, there was no statistically significant difference between intervention groups in facility delivery and contraceptive use. However, both one-way and two-way text messages improved early breastfeeding compared to standard care (week 10: 79%$_{control}$ versus 93%$_{one-way text}$ versus 96% $_{two-way text message}$) [25]. Further, one cluster randomized trial investigated the effect of

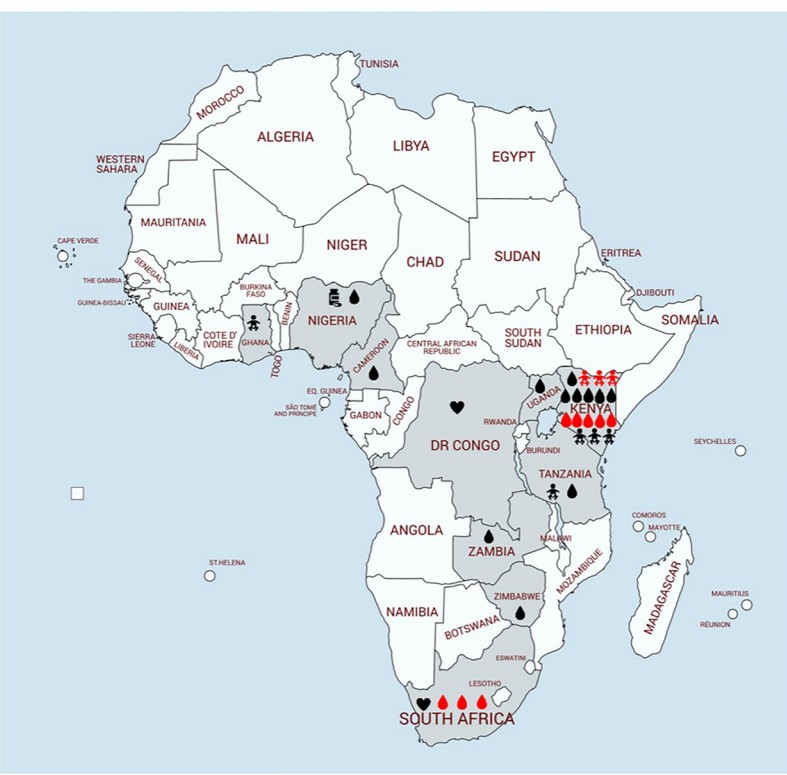

| Clinical area ＼ Trial status | HIV | Malaria | Reproductive/ Antenatal health* | NCDs† |
|---|---|---|---|---|
| Published trial (n=20) | ● | ▣ | �astr | ♥ |
| Unpublished trial (n=11) | ● | ▣ | ☆ | ♥ |

*Reproductive/antenatal health: Reproductive/antenatal knowledge, postpartum contraception, antenatal care visit, facility delivery, early breast feeding.
†Non-communicable diseases: Diabetes, hypertension.
Republished from mapchart.net under a CC BY license, with permission from Minas Giannekas, original copyright 2021.

**Fig 2. Overview of two-way text message trials in Africa according to clinical area and trial status.**

two-way text messages that promoted contraception use and compared these to a control group receiving a sham text message (i.e. text messages with nutrition-focused-content) on the incidence of unintended pregnancies among sex workers in Kenya. They found no difference in the incidence of unintended pregnancy over 12 months of follow-up in the intervention group compared to the control group (hazard ratio (HR): 0.98; 95% CI: 0.7 to 1.4) [35]."

Four trials reported outcomes related to HIV. One trial investigated the effect of two-way text messages (intervention) compared to routine in-person visits (standard care) for post-operative follow-up among male circumcision patients Zimbabwe and did not find any significant difference in adverse events between groups (risk difference (RD): 1.04% (p = 0.32)) [22]. Another trial reported that a two-way text-message intervention did not improve HIV medicine adherence (measured as continuous outcome) in Uganda compared to controls (mean proportion of pills taken$_{two-way}$: 0.64 versus mean$_{control}$: 0.67 (p = 0.15) [27]. Further, two trials assessed HIV medicine adherence and appointment attendance dichotomously but did not report the required patient and event data for be included in the meta-analyses. One of these trials was an unpublished trial set in South Africa (results recorded in clinicaltrials.gov) found a marginally effect in the two-way text message group compared to the control group (OR: 1.9; 95% CI: 0.9 to 4.2) [41]. The other trial was 3-arm trial on prevention of mother-to-child transmission of HIV in Kenya [44]. The trial found that two-way text messages did not improve on-

time clinic appointment attendance (aRR$_{\text{two-way compared to control}}$: 1.01; 95% CI: 0.98 to 1.04) and medicine adherence (measured as viral load non-suppression at any time) compared to a one-way text message group or a control group (aRR$_{\text{two-way compared to control}}$: 0.80; 95% CI:0.52 to 1.23) [44].

## Risk of bias assessment

A total of twelve trials were assessed for risk of bias as they were included in either the primary meta-analyses or sensitivity analyses; ten trials were included in the primary meta-analyses and two additional trials was included in a sensitivity or subgroup analysis. Three trials [26, 30, 43] were judged as overall low risk of bias, and nine as overall high risk of bias [19–21, 23, 24, 28, 29, 31, 45] (Fig 3, S5 File). Six out of twelve trials had unclear risk of selection bias due to their method of randomization or allocation concealment being inadequately described [19–21, 23, 28, 29]. Further, three trials had high risk of reporting bias as the information in trial registries were recorded after the trial had started or because the trial did not report all pre-specified outcomes [21, 28, 31].

## Meta-analysis

**Primary analyses.** Sixteen published trials reported results on either medicine adherence and/or appointment attendance and of these, ten trials were included in the primary meta-analyses. Further, one cluster trial did not adjust for clustering and were excluded from the primary analysis on appointment attendance, though included in a subsequent sensitivity analysis [20], and one trial only compared two-way to one-way messages and was included in another sensitivity analysis [43]. One trial was excluded from the meta-analysis as it only reported continuous data on "medicine adherence" [32] and three trials did not report results in a format that allowed inclusion in meta-analysis [27, 41, 44]. Among the included trials, one trial had a 3-arm intervention, where two arms had two different types of two-way text messages and one control arm. In the meta-analysis, we used the results from the tailored two-way text message arm, where the messages targeted the participants self-reported stages of change according to a theory of change model [28]. Further, one trial reported data on both appointment attendance and medicine adherence and were included in both analyses [23].

Five trials (6627 participants) were included in our primary analysis on appointment attendance [19, 23, 24, 26, 28]. Overall, two-way text messages did not improve appointment attendance compared with standard care on appointment attendance, RR: 1.03; 95% CI: 0.95–1.12, $I^2$ = 53% (Fig 4) (S3 File).

Six trials (3362 participants) were included in our primary analysis on medicine adherence [21, 23, 29–31, 45]. Overall, two-way text messages improved medicine adherence compared to standard care, RR: 1.14; 95% CI: 1.07–1.21, $I^2$ = 8% (Fig 5).

**Subgroup analyses.** When comparing high risk to low risk of bias trials for appointment attendance, we found that high risk of bias trials reported a seemingly higher effect of two-way interventions compared to the single low risk of bias trial, though the subgroup difference was not statistically significant; RR$_{\text{high risk}}$: 1.10; 95% CI: 0.95–1.28 versus RR$_{\text{low risk}}$: 0.98; 95% CI: 0.91–1.05 (interaction test p = 0.15) (Fig C in S4 File). Similarly, for medicine adherence, we found a seemingly increased intervention effect when comparing high risk of bias trials with the single low risk of bias trial, though the difference in effect was not statistically significant: RR$_{\text{high risk}}$: 1.16, 95% CI: 1.08–1.23 versus RR$_{\text{low risk}}$: 1.02; 95% CI: 0.85–1.22 (interaction test: p = 0.21) (Fig D in S4 File).

When stratifying the primary analyses into different clinical areas, eight out of ten trials concerned HIV care [19, 21, 24, 26, 28, 30, 31, 45] whilst the remaining two trails concerned

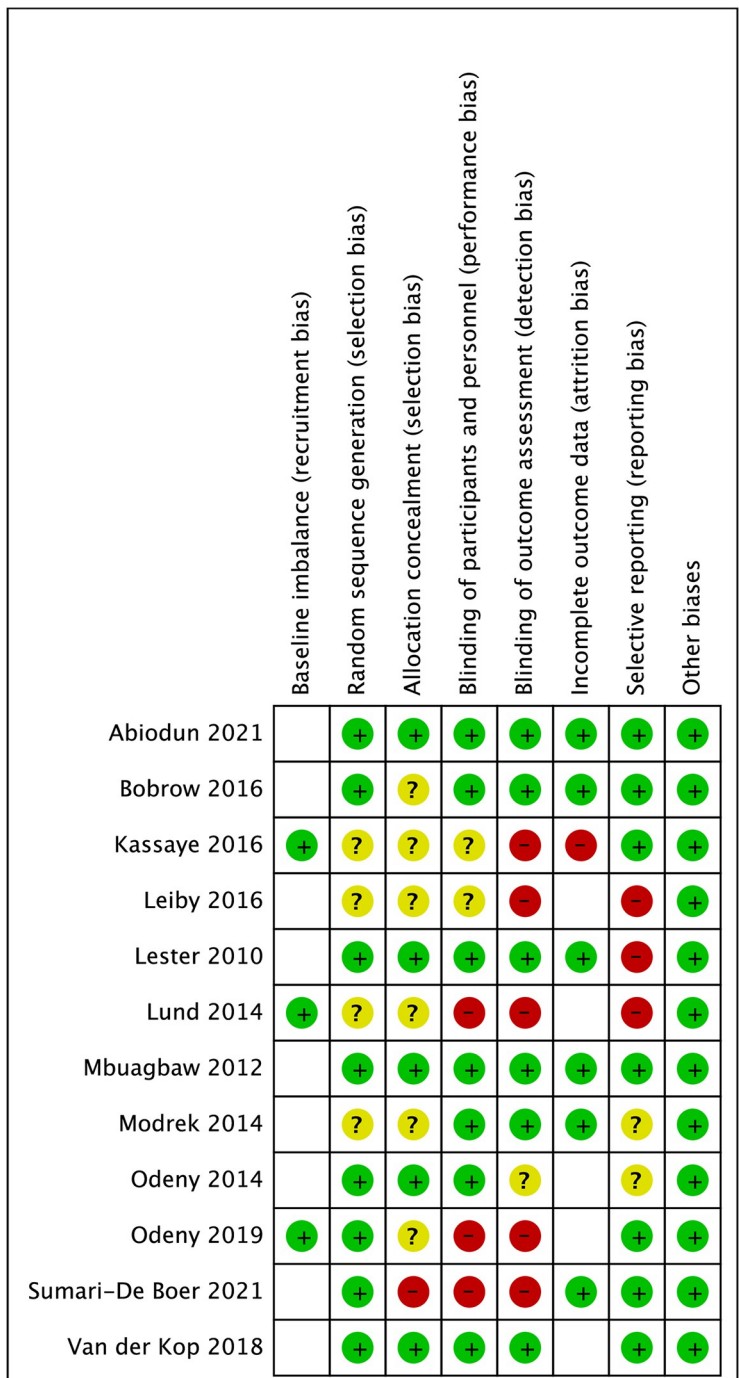

**Fig 3. Risk of bias assessment**\*. \*Empty cell: No risk of bias assessment—Attrition bias was not assessed for trials that only reported "appointment attendance" as an outcome as incomplete data is an integral part of this outcome.

hypertension [23] and malaria [21]. When comparing the various clinical areas, there was no significant differences for neither appointment attendance ($RR_{hypertension}$: 0.98, 95% CI 0.93 to 1.04 versus $RR_{HIV}$: 1.10, 95% CI 0.95 to 1.29) or medicine adherence ($RR_{hypertension}$: 1.18, 95% CI 1.02 to 1.36 versus $RR_{HIV}$: 1.09, 95% CI 1.01 to 1.19 versus $RR_{malaria}$: 1.22, 95% CI 1.08 to 1.37) (Fig E, F in S4 File).

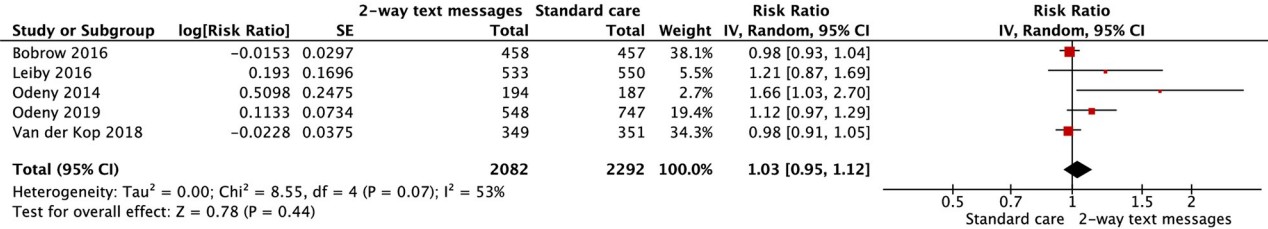

**Fig 4. Effect of two-way text messages versus standard care on appointment attendance.**

When comparing the different types of two-way text messages, we found a seemingly higher intervention effect on appointment attendance in trials where participants had the possibility to get in contact with a HCP through a phone call, though the difference in effect was not significant, $RR_{HCP}$ contact: 1.27; 95% CI: 0.87–1.83 versus $RR_{HCP}$ no contact: 0.99; 95% CI: 0.94–1.03 (interaction test p = 0.17) (Fig G in S4 File). For medicine adherence, it appeared that the text option was more effective than the phone call option, $RR_{HCP}$ contact: 1.03; 95% CI: 0.93–1.15 versus $RR_{No\ HCP}$ contact: 1.19; 95% CI: 1.11–1.29 (interaction test: p = 0.03) (Fig H in S4 File).

## Sensitivity analyses

One cluster trial only reported results unadjusted for clustering [20] and was included in a subsequent sensitivity analysis for appointment attendance. This analysis showed an increased intervention effect compared to our primary analysis but with an increase in heterogeneity, RR: 1.15; 95% CI: 0.99–1.33, $I^2$ = 89% (Fig I in S4 File). When excluding both cluster trials in a sensitivity analysis for appointment attendance, the intervention effect of two-way text messages on appointment attendance did not change from our primary analysis, $RR_{cluster\ excluded}$: 1.01; 95% CI 0.93–1.10, $I^2$ = 50% (Fig J in S4 File). For medicine adherence, we found that when excluding the single cluster trial the intervention effect was still statistically significant, $RR_{cluster\ excluded}$: 1.17; 95% CI: 1.09–1.25, $I^2$ = 0% (Fig K in S4 File).

When using a fixed effect model in our sensitivity analysis for appointment attendance and medicine adherence, the estimate was quite similar to the results from the random effects model, $RR_{fixed,\ appointment\ atttendance}$: 1.00; 95% CI: 0.96–1.05, $I^2$ = 53% (Fig L in S4 File); $RR_{fixed,\ medicine\ adherence}$: 1.14; 95% CI: 1.07–1.21, $I^2$ = 8% (Fig M in S4 File).

Only one trial compared two-way messages to one-way messages for appointment attendance [23] and two trials compared two-way messages to one-way messages for medicine adherence [23, 43]. There was a tendency for one-way messages to be more effective than two-way messages for medicine adherence, RR: 0.89; 95% CI: 0.80–1.00, Fig B in S4 File).

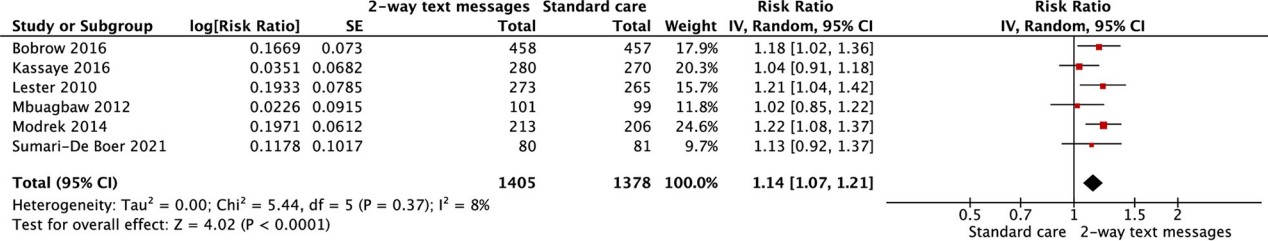

**Fig 5. Effect of two-way text messages versus standard care on medicine adherence.**

## Discussion

### Main findings

In this systematic review and meta-analysis of two-way text message trials set in Africa, we identified a total of 31 trials; 20 published and 11 unpublished trials, of which 10 were eligible to be included in our primary meta-analyses as they reported binary data on appointment attendance or medicine adherence. Our analyses showed an effect of two-way messages on medicine adherence and an uncertain effect on appointment attendance, but most trials had high risk of bias. Our sensitivity and subgroup analyses did not change the overall conclusions of our primary analyses. However, when comparing the effect estimates for both appointment attendance and medicine adherence in relation to risk of bias, we found a seemingly higher intervention effect in trials with high risk of bias compared to low risk of bias trials though the subgroup differences were not statistically significant. Further, most trials included in the meta-analyses concerned HIV (n = 8/10), hence, the findings mainly relate to this field. Overall, our descriptive analysis showed that two-way text messages slightly improved some clinical outcomes including reproductive health knowledge, post-partum contraceptive use, and early breastfeeding.

### Strength and limitations

To our knowledge, this is the first systematic review and meta-analysis that specifically investigates the effect of two-way text message interventions in an African setting, where the receiver can interact with the sender. Our analyses demonstrate that the effect of two-way text messages varies within different outcomes, clinical areas, and types of interventions and provides an in-depth perspective on effect of such mHealth interventions in an African context. A strength of this review is that it is based on a predefined protocol and that we conducted a comprehensive search that involved both published and unpublished trials. However, the review also has several limitations. A two-way text message intervention is a rather simple, low-cost intervention that overall appears effective for medicine adherence in an African context. However, participants must have access to a mobile phone and be able to read and send text messages or have family members who are able to. The illiteracy rate among adults in Sub-Saharan Africa was 35% in 2019, and such individuals may not benefit from the interventions to the same extent as literates [51]. Nevertheless, it is not given that people deemed illiterate in a classic reading and writing context cannot benefit from text message interventions as they may still be capable of understanding text messages. Furthermore, we only identified trials from 10 out of 54 countries in Africa, and the majority (17 of 31) were set in Kenya. This may limit the generalizability of our results to the whole of Africa as countries are heterogenous in relation to culture, health care systems, and socio-economic status. Furthermore, there are huge domestic differences within African countries. Future researchers should take demography and national geography into account to get a better understanding of who two-way text messages benefits the most and how to develop a mHealth tool that reaches as many as possible.

We included all types of health outcomes in this review and conducted meta-analyses on the outcomes "appointment attendance" and "medicine adherence" whilst other outcomes were described descriptively. The majority of included trials reported results on either medicine adherence or appointment attendance, yet this still only entailed that 10 trials were included our meta-analyses as the other were reported in a format that did not allow for inclusion. Further, eight out of ten trials included in the meta-analyses concerned "HIV". This limits the generalizability of our findings beyond the field of HIV and reduce the potential impact of this review on public health practice. As evidence from African mHealth trials within

different clinical areas is growing, we recommend that reviews like this one are regularly undertaken. This will allow for additional meta-analyses within different clinical areas, which will provide a more nuanced insight into the effect of text message trials across clinical fields and increase the impact on public health practice.

Only two out of ten trials included in our primary meta-analyses were judged to be in overall low risk of bias [26, 30] whilst eight had overall high risk of bias [19, 21, 23, 24, 28, 29, 31, 45]. Hence, the quality of trials was generally low, and this may have biased the effect estimates which was also somewhat indicated in our subgroup analysis. Further, we only focused on intervention effects in relation to two health outcomes of interest and we did not focus on other aspects of mHealth interventions, such as the participants perspectives. As suggested by the WHO guide "monitoring and evaluating digital health interventions" and the model for assessment of telemedical applications (MAST), an evaluation of not only the clinical effectiveness but also other dimensions, which the technology affects, may provide a more nuanced understanding of how technological interventions truly work [51, 52].

Finally, another limitation of this review is that two-way text messages can be considered a heterogenous type of intervention, and one could question whether it is fair to pool them as one type of intervention as done in our primary meta-analyses. It is plausible that an intervention where you achieve direct contact with a health care professional and an intervention where you have the option to answer a quiz, may have a different effect. We tried to address the issue of comparability in our subgroup analyses. However, for future reviews, researchers should aim to segregate the different types of two-way text message interventions to get a better understanding of the true impact of various types of two-way text messages.

## Comparison of findings

HIV was the most prevalent clinical condition in in our meta-analyses, and in line with our findings a systematic review of one-way text message interventions in Africa found a fairly similar effect on HIV medicine adherence (RR: 1.18, 95% CI: 1.02 to 1.37) [5]. This indicates that different types of text message interventions–both those that allow for interaction with the sender and those that do not–may improve HIV medicine adherence in an African context. This may be due the interventions lowering the threshold for contact with health care personnel and clinics and simultaneously works as a supportive and reminder tool. A systematic review and meta-analysis from 2019, which investigated the effectiveness of mobile phone interventions and HIV medicine adherence, found a somewhat similar effect for text message interventions, though not significant and with high heterogeneity (RR: 1.25; 95% CI: 0.97 to 1.61; $I^2$ = 74%) [53]. The trials in that review were conducted globally and included high-, low- and middle-income countries. It is plausible that the effect of text messages on HIV medicine adherence may differ across various settings, and more literature comparing the effect of mHealth interventions in high- and low-income countries are needed.

When looking at one-way versus two-way text messages on medicine adherence we found a tendency for one-way text messages to more effective than two-way, though the finding was not significant and only based on two trials (RR: 0.89, 95% CI: 0.8–1.0). In contrast, a systematic review and meta-analysis from 2015 found a higher intervention effect of two-way text messages compared to one-way text messages on medicine adherence ($RR_{two-way:}$ 1.23 95% CI: 1.12–1.35; $RR_{one-way}$:1.04 95% CI: 0.97–1.11, p = 0.007), yet this review did not have any geographical restrictions [6]. When comparing our findings to the systematic review and meta-analysis that landscaped the effect of one-way text messages trials in an African setting [5], the other review found that one-way text messages improved appointment attendance (OR: 2.03; 95% CI: 1.40 to 2.95) though not medicine adherence (RR: 1.10; 95% CI: 0.98 to 1.23), whilst

we found an effect of two-way text messages on medicine adherence and an uncertain effect on appointment attendance. Yet we had few data in this review, which may have influenced our results and the apparent differences in results could be due to lack of statistical power. Therefore, more large high-quality trials are needed, preferably three arm trials, to test if there is a difference in effect between one-way and two-way text message interventions in an African context.

WHO´s strategy on digital health 2020–2024 emphasizes the importance of building evidence, developing, and disseminating knowledge on eHealth in order to help policymakers and public servants to understand the power and complexity of eHealth. Their vision is that appropriate digital health technologies is a key component to achieving health for all [54]. This review has shown that one element of digital health—in the form of two-way text messages– have potential to improve medicine adherence in an African context. Yet our findings and our comparison of findings to other reviews also shows that text message interventions are complex, and the effect of such interventions is not uniform and easy to interpret. Hence, to better understand the clear effect of text message interventions, it is important that future studies clearly define the type of text message intervention they are undertaking and compare its effect to other similar interventions. We recommend "The mHealth evidence reporting and assessment checklist" as a reporting guide for future trials, to support replication and improve the quality of mobile phone health intervention trials [4].

## Conclusions

Two-way text messages seemingly improve medicine adherence but has an uncertain effect on appointment attendance and other healthcare outcomes compared to standard care. We recommend taking this into account when planning future digital health strategies in Africa. To better estimate the true impact of two-way text message interventions, we recommend that future trials and reviews also consider other aspects of technological interventions, such as feasibility, usability and sustainability as recommended by the WHO.

## Supporting information

**S1 File. PRISMA checklist.**
(PDF)

**S2 File. Database search strategy.**
(PDF)

**S3 File. Dataset for meta-analysis.**
(PDF)

**S4 File. Subgroup and sensitivity analyses.**
(PDF)

**S5 File. Risk of bias assessment.**
(PDF)

## Acknowledgments

We are grateful for the assistance of Herdis Foverskov and Mette Brandt Eriksen, from the library at the University of Southern Denmark, who assisted in developing the search strategies.

## Author Contributions

**Conceptualization:** Andreas Lundh, Ditte S. Linde.

**Data curation:** Emilie S. Ødegård, Lena S. Langbråten.

**Formal analysis:** Emilie S. Ødegård, Lena S. Langbråten, Ditte S. Linde.

**Methodology:** Emilie S. Ødegård, Lena S. Langbråten, Andreas Lundh.

**Supervision:** Andreas Lundh, Ditte S. Linde.

**Validation:** Andreas Lundh, Ditte S. Linde.

**Visualization:** Ditte S. Linde.

**Writing – original draft:** Emilie S. Ødegård, Lena S. Langbråten.

**Writing – review & editing:** Andreas Lundh, Ditte S. Linde.

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
