## [Decision Letter · Decision Letter 0]

4 Aug 2021

PONE-D-21-10098

Two-way text message interventions and healthcare outcomes in Africa: Systematic review of randomized trials with meta-analysis

PLOS ONE

Dear Dr. Linde,

Thank you for submitting your manuscript to PLOS ONE. After careful consideration, we feel that it has merit but does not fully meet PLOS ONE’s publication criteria as it currently stands. Therefore, we invite you to submit a revised version of the manuscript that addresses the points raised during the review process.

We look forward to receiving your revised manuscript.

Kind regards,

Michelle L. Munro-Kramer, PhD, CNM, FNP-BC

Academic Editor

PLOS ONE

Additional Editor Comments (if provided):

Thank you for this comprehensive manuscript. Both reviewers have provided detailed comments that should be addressed on resubmission. Please ensure that all reference numbers and references are accurate before resubmission.

Journal Requirements:

3. Please update search and analysis to include studies published since March 2020

5. "Please upload a new copy of Figures 1,2,4 and 5 as the detail is not clear. Please follow the link for more information: https://blogs.plos.org/plos/2019/06/looking-good-tips-for-creating-your-plos-figures-graphics/" https://blogs.plos.org/plos/2019/06/looking-good-tips-for-creating-your-plos-figures-graphics/.

6. We note that Figure 2 in your submission contain map images which may be copyrighted. All PLOS content is published under the Creative Commons Attribution License (CC BY 4.0), which means that the manuscript, images, and Supporting Information files will be freely available online, and any third party is permitted to access, download, copy, distribute, and use these materials in any way, even commercially, with proper attribution. For these reasons, we cannot publish previously copyrighted maps or satellite images created using proprietary data, such as Google software (Google Maps, Street View, and Earth). For more information, see our copyright guidelines: http://journals.plos.org/plosone/s/licenses-and-copyright.

Reviewers' comments:

Reviewer's Responses to Questions

**Comments to the Author**

1. Is the manuscript technically sound, and do the data support the conclusions?

Reviewer #1: Partly

Reviewer #2: Yes

2. Has the statistical analysis been performed appropriately and rigorously? 

Reviewer #1: Yes

Reviewer #2: Yes

3. Have the authors made all data underlying the findings in their manuscript fully available?

Reviewer #1: Yes

Reviewer #2: Yes

4. Is the manuscript presented in an intelligible fashion and written in standard English?

Reviewer #1: Yes

Reviewer #2: Yes

5. Review Comments to the Author

Reviewer #1: 1. Overall

1.1. I find this article to be a robust and interesting manuscript. However, there is a major methodological question of the definition of a “two-way” text messaging intervention versus a one-way intervention. Several of the interventions included in this review - that meet the eligibility criteria the authors used - I would not necessarily describe as two-way. For example, Ref 33 provides an option to participants to text in for more electronic information (delivered electronically and via automation) but this is only an option and participants may or may not use it; many of the other included studies could be similarly described. Other included studies have what could be considered a weak two-way option where a simple quiz may be occasionally used. I find this a major weakness of this manuscript and the authors need to speak to this more extensively throughout the paper.

1.2 Furthermore, I think it would be a stronger methodology to focus on two-way text messaging interventions that provide the option to speak with a health care provider, rather than the option to electronically access additional information which is common in the studies included in this review. Perhaps this could be used to describe the type of two-way text messaging intervention, and then use this variable to test for differences in effects based on automated replies versus reply from a live health care provider.

2.0 Comments on Specific Sections of the Manuscript

2.1 Introduction

2.1.1 Page 3 Line 67, please elaborate on what you mean by “various types of text message interventions.” Also you refer to “type of intervention” and use this as a comparison criteria / potential moderator throughout the paper, so please describe what is meant here and how your classification scheme was developed for this analysis.

2.1.2 Page 4 Line 75, please elaborate on “other health outcomes”; furthermore, please justify why appointment and medication adherence were the two focal outcomes when there are so many relevant and important healthcare outcomes to look at

2.1.3 The introduction would benefit from discussion of why two-way text messaging may be better than one-way. Adding theory and evidence behind this hypothesis would strengthen the introduction, which is very short.

2.2 Methods

2.2.1 Page 4 Eligibility Criteria, was any outcome eligible for inclusion as long as it met your other eligibility criteria? Please clarify and explain this better and discuss this in your Introduction

2.2.2 Eligibility Criteria, Please describe how (and with what search terms) two-way text message interventions were included in your search strategy; often you cannot tell whether an SMS intervention is 1 or 2 way unless a careful review of the intervention/report is undertaken. This should also be mentioned as a limitation of your review, in that it’s likely 2-way interventions were missed and others might describe two-way interventions differently than the authors did for this manuscript.

2.2.3 Page 5 Line 122, were all trials assessed for bias by two people - so 100% double coding for risk of bias? This is not clear from the text.

2.3 Results

2.3.1 Page 12 Descriptive Analyses, i think this paragraph would benefit from a revised subtitle and explanation of what is being presented in this section. Initially it seemed like it would be trials not included in other analyses, but this is not the case. Please clarify and revise.

2.4 Discussion

2.4.1 Page 17 Line 351, I do not think that it is accurate to state that people who are deemed “illiterate” in some contexts would not benefit from text messages and interventions; we have found that people can be literate in some information contexts while “illiterate” in others - and that even people who are deemed “illiterate” in classic reading/writing literacy can have a high capability of understanding text messages.

2.4.2 Page 20 Line 410-411, I would suggest including a reference to the mERA checklist (BMJ 2016;352:i1174 http://dx.doi.org/10.1136/bmj.i1174) - this checklist contributes to standardization in reporting/comparisons between interventions, and ensuring that essential aspects of mHealth interventions are attended to.

2.5 Overall more description and discussion of two-way text messaging is needed throughout the paper.

Reviewer #2: This is a timely review of an important tool in the mobile health toolbox: two-way texting interventions for improved health outcomes. The paper is well written and well organized, as is expected from this highly qualified author group. It is also critical that these reviews, although not novel, are completed routinely as the evidence is growing and the technology changing rapidly.

However, the included studies are few and the outcomes limited, reducing the potential impact of this review on public health practice. There are many reviews of texting interventions – as the authors note and cite – including many focused on HIV, as is the majority of this one. Could more health topics be included (malnutrition, diabetes, vaccinations, etc)? Could the study assessment criteria extend beyond a more simple assessment of high or low bias to a more comprehensive and nuanced review of the study rigor?

Furthermore, the call for more better evidence has been made for over a decade. The authors know and could give more specific suggestions on what is needed. For example, the authors could strengthen the paper by making more detailed suggestions for what could create an improved evidence base, potentially including focal areas in the WHO guide, “Monitoring and evaluating digital health interventions: A practical guide to conducting research and assessment” (https://www.who.int/reproductivehealth/publications/mhealth/digital-health-interventions/en/ ). Whether referring to that guide or other evaluation criteria, the authors could make a larger contribution to the digital health community if the review would focus more on the weaknesses of the included studies (or the excluded studies), moving past a more generic call for “more high quality trials.” That would exponentially increase the impact of this paper.

Other comments:

1. The words used for searching seem too exclusive. Other than, “appointment attendance” and “medicine adherence”, what other words were used? Strings like “visit attendance,” “treatment adherence,” and “linkage to care” might have also brought in trials. Were these key words also considered? Texting, cell phones or digital health may have brought up other studies.

2. Lines 49-51 and 51-53: break into 2 sentences for clarity.

3. Many sentences in the intro would benefit from a grammatical review.

4. Line 85: males and females separate or also together?

5. Line 150: what about for non dichotomous outcomes, i.e. % of on-time visits/year, for example?

6. 154: what is I2 ? What is it or spell out.

7. Line 185: This does not make sense. How can infants be the target group for a mHealth innovation? Does this mean the guardian or parent of an infant?

8. Line 213: This specific RCT was testing whether post-operative follow-up by two-way text messages was as safe as in-person visits for follow-up to identify and report adverse events. The intent was to use two-way messaging as a form of telehealth – comparing adverse events between the intervention arm as compared to standard care – trying to reduce workload of nurses. Two-way texting actually identified /more/ adverse events (a positive for quality care) as the men were provided with reassurance on wound care and encouraged to return to care if they had a problem. Two-way texting was not non-inferior (the texting actually improved reporting), and the lack of significant difference between the arms is a positive outcome for potential workload reduction and quality care. The original sentence starting on line 213, “Another trial investigated the effect of two-way text message intervention on circumcision harms in Zimbabwe and did not find any differences compared with standard care (risk difference (RD): 1.04% (p=0.32) [20],” is incorrect and misleading. Texting was not related to harms but on identifying potential harms. It should be replaced with something like, “Another trial investigated the effect of two-way text messages (intervention) as compared to routine in-person reviews (standard care) for post-operative follow-up among male circumcision patients in Zimbabwe and did not find any significant difference in adverse events between arms (risk difference (RD): 1.04% (p=0.32) [20].

However, although this study outcome [20] was not among those studies included in the primary outcomes of the overall meta-analysis, misunderstanding these study outcomes does give pause as to the veracity of the other findings reported in the paper. I am not going to review the content of each included paper, but the authors should be confident that the content of the meta-analysis and the study summaries are correct.

9. Lines 225-227, what is the outcome of the study? The sentence tells only what the study investigated.

10. Line 289: so all HIV studies were compared to the one hypertension study? What would a significant finding tell the reader anyway if almost all trials were for HIV?

11. Line 298. What is meant by the effect of the call back versus text back option mean? Is this comparing whether the clients sent back an SMS or called back to confirm attendance? To confirm attendance intent? Please clarify.

12. Line 300: please clarify again what is meant by call back vs. SMS. This is critical to readers’ understanding of what these results may mean for future interventions. Does this mean giving clients a chance to call back was more effective for medicine adherence?

13. Lines 334-338 lack clarity, “compared to standard care, two-way text messages slightly improved diabetes control [32], reproductive health knowledge [31, 39] post-partum contraceptive use [30], early breastfeeding [23] though not adverse events [22], unintended pregnancy [33] or HIV medicine adherence [42].

I /think/ that citation [22] here should be [20]. First, this makes me request that you review all your citations within the paper. Second, if [22] is meant to be [20], please rephrase to, ….”early breastfeeding [23] and adverse event ascertainment [20] although not unintended pregnancy [33] or HIV medicine adherence [42].” However, isn’t this metareview about HIV medicine adherence? Why wasn’t [42] included above?

14. Line 359: the restricted outcomes to only 2 (appointment attendance or medicine adherence) is also a large limitation. What other outcomes are the intended impact of two-way texting interventions? Testing uptake? Linkage to care? Self-monitoring? Smoking or alcohol harms reduction? Intimate partner violence? Improved nutrition or malnutrition identification?

15. Line 365: text back options are not well explained. What does this mean? I thought that the previous lines (298-302) noted the impact of the call back? This is unclear, muddling a potential impact of this type of review on actual intervention development.

16. Line 402: Spell out WHO the first time

6. PLOS authors have the option to publish the peer review history of their article (what does this mean?). If published, this will include your full peer review and any attached files.

Reviewer #1: No

Reviewer #2: No

---

## [Author Response · Author response to Decision Letter 0]

4 Jan 2022

Journal Requirements

Comment Response

#1. Please ensure that your manuscript meets PLOS ONE's style requirements, including those for file naming. 

Response #1: Thank you for pinpointing this. We have gone through all documents to ensure they are aligned with PLOS ONE’s requirements.

#2. Please review your reference list to ensure that it is complete and correct. If you have cited papers that have been retracted, please include the rationale for doing so in the manuscript text, or remove these references and replace them with relevant current references. Any changes to the reference list should be mentioned in the rebuttal letter that accompanies your revised manuscript. If you need to cite a retracted article, indicate the article’s retracted status in the References list and also include a citation and full reference for the retraction notice. 

Response #2: We have reviewed our reference list and updated it with new references, which were found during our updated literature search.

3. Please update search and analysis to include studies published since March 2020 

Response #3: Thank you for this comment. We have updated search and included three new trials to the review:

Abiodun O, Ladi-Akinyemi B, Olu-Abiodun O, Sotunsa J, Bamidele F, Adepoju A, et al. A Single-Blind, Parallel Design RCT to Assess the Effectiveness of SMS Reminders in Improving ART Adherence Among Adolescents Living with HIV (STARTA Trial). J Adolesc Health. 2021;68:728-36. doi: 10.1016/j.jadohealth.2020.11.016. 

Kinuthia J, Ronen K, Unger JA, Jiang W, Matemo D, Perrier T, et al. SMS messaging to improve retention and viral suppression in prevention of mother-to-child HIV transmission (PMTCT) programs in Kenya: A 3-arm randomized clinical trial. PLoS Med. 2021;18:e1003650. doi: 10.1371/journal.pmed.1003650.

Sumari-de Boer IM, Ngowi KM, Sonda TB, Pima FM, Masika Bpharm L, Sprangers MAG, et al. Effect of Digital Adherence Tools on Adherence to Antiretroviral Treatment Among Adults Living With HIV in Kilimanjaro, Tanzania: A Randomized Controlled Trial. J Acquir Immune Defic Syndr. 2021;87:1136–44. doi: 10.1097/QAI.0000000000002695,

#4. We note that you have indicated that data from this study are available upon request. PLOS only allows data to be available upon request if there are legal or ethical restrictions on sharing data publicly. For more information on unacceptable data access restrictions, please see http://journals.plos.org/plosone/s/data-availability#loc-unacceptable-data-access-restrictions. 

Response #4: Thank you for pinpointing this error. There is no restriction for data availability. We have attached an anonymized data set as “Supporting Information File 6” and updated our data availability statement.

#5. Please upload a new copy of Figures 1,2,4 and 5 as the detail is not clear 

Response #5: We have uploaded new copies of figures 1-5. 

#6. We note that Figure 2 in your submission contain map images which may be copyrighted. All PLOS content is published under the Creative Commons Attribution License (CC BY 4.0), which means that the manuscript, images, and Supporting Information files will be freely available online, and any third party is permitted to access, download, copy, distribute, and use these materials in any way, even commercially, with proper attribution. For these reasons, we cannot publish previously copyrighted maps or satellite images created using proprietary data, such as Google software (Google Maps, Street View, and Earth). … 

Response #6: Thank you for pinpointing this issue. We have contacted MapChart and been granted permission to use the map and have attached the signed permission form as documentationBelow the figure it now states:

“Reprinted from mapchart.net under a CC BY license, with permission from Minas Giannekas, original copyright 2021.” 

Reviewer 1

Comment

#1.1. I find this article to be a robust and interesting manuscript. However, there is a major methodological question of the definition of a “two-way” text messaging intervention versus a one-way intervention. Several of the interventions included in this review - that meet the eligibility criteria the authors used - I would not necessarily describe as two-way. For example, Ref 33 provides an option to participants to text in for more electronic information (delivered electronically and via automation) but this is only an option and participants may or may not use it; many of the other included studies could be similarly described. Other included studies have what could be considered a weak two-way option where a simple quiz may be occasionally used. I find this a major weakness of this manuscript and the authors need to speak to this more extensively throughout the paper. 

Response #1.1: We thank the reviewer for the constructive comment. We agree with the reviewer that we have a broad definition of two-way messages, which could clutter the true effect of the stronger two-way text message options. Yet we still find that the distinction between having the option to interact with the sender through two-way options compared to only receiving one-directional messages is important clarify. The reasons for this broad definition was that 1) we wanted to provide an overview of the field 2) we were afraid that more narrow inclusion criteria could results in loss of power in our meta-analyses. We agree that we have not clarified the limitation of using broad inclusion criteria clearly in the manuscript and added the following paragraph to the discussion:

“Finally, another limitation of this review is that two-way text messages can be considered a heterogenous type of intervention, and one could question whether it is fair to pool them as one type of intervention as done in our primary meta-analyses. It is plausible that an intervention where you achieve direct contact with a health care professional and an intervention where you have the option to answer a quiz, may have a different effect. We tried to address the issue of comparability in our subgroup analyses. However, for future reviews, researchers should aim to segregate the different types of two-way text message interventions to get a better understanding of the true impact of various types of two-way text messages.”

#1.2 Furthermore, I think it would be a stronger methodology to focus on two-way text messaging interventions that provide the option to speak with a health care provider, rather than the option to electronically access additional information which is common in the studies included in this review. Perhaps this could be used to describe the type of two-way text messaging intervention, and then use this variable to test for differences in effects based on automated replies versus reply from a live health care provider. 

Response #1.2: We thank the reviewer for this suggestion. We have rephrased the description of our subgroup analyses; however, we have not changed our meta-analyses as they were conducted according to our pre-defined protocol. The methodology section now states:

“Subgroup analyses were performed comparing overall low risk of bias trials with high risk of bias trials, clinical areas and types of interventions, e.g. if the receiver could interact with a health care professional (HCP) through a phone call or text message.”

#2.1: Introduction: 2.1.1 Page 3 Line 67, please elaborate on what you mean by “various types of text message interventions.” Also you refer to “type of intervention” and use this as a comparison criteria / potential moderator throughout the paper, so please describe what is meant here and how your classification scheme was developed for this analysis. 

Response #2.1.1: We thank the reviewer for this comment. Further down in our introduction we state what we mean by the different interventions. We hope this answers the reviewer’s question:

“Further, a series of Cochrane reviews published between 2012 and 2017 assessed the effect of various text message interventions on different health issues [7-12], but the reviews did not differentiate between one-way and two-way text message interventions in their analyses, hence, the effect of one-way versus two-way text messages interventions was unclear.

As stated in our response to comment #1.2, we have clarified what we mean by “type of intervention”, and stated in our methodology section that it means whether the receiver could interact with a health care professional.

#2.1.2 Page 4 Line 75, please elaborate on “other health outcomes”; furthermore, please justify why appointment and medication adherence were the two focal outcomes when there are so many relevant and important healthcare outcomes to look at 

Response #2.12: We thank the reviewer for this comment and have elaborated on our choice of outcomes. The following is now stated in the introduction:

“Research shows that linkage to care is a challenge in many resource-limited settings [13], and it is plausible that two-way text messages – which allow for interactive communication with health care providers – may be a more effective health care tool than one-way messages. Appointment attendance and medicine adherence are outcomes frequently used to measure linkage to care across clinical settings, yet the effect of two-way messages on these outcomes in African context is unclear. Therefore, this systematic review aims to landscape randomized trials in Africa involving two-way text message interventions compared to standard care or one-way text messages and analyze their effect on appointment attendance and medicine adherence. By conducting a meta-analysis on the same outcomes as the previous systematic review of one-way messages in Africa [5], this review will provide an even greater understanding of how various types of text message interventions work in an African context. Further, in line with the previous review, we also describe other health outcomes to provide a full overview of two-way message trials set in Africa.”

#2.1.3 The introduction would benefit from discussion of why two-way text messaging may be better than one-way. Adding theory and evidence behind this hypothesis would strengthen the introduction, which is very short.

Response #2.1.3: Thank for this suggestion. We have taken this comment into consideration and updated our introduction. It now states:

“A systematic review and meta-analysis from 2015 – which included trials conducted in various settings – suggested that interactive communication between a client and a healthcare provider may lead to greater support, motivation, and safety for the patient. The meta-analysis found a higher intervention effect of two-way messages compared to one-way text messages on medicine adherence [6].” 

#2.2 Methods: 2.2.1 Page 4 Eligibility Criteria, was any outcome eligible for inclusion as long as it met your other eligibility criteria? Please clarify and explain this better and discuss this in your Introduction

Response #2.2.1: Trials reporting all health care outcomes were included if they met all other eligibility criteria. This has now been clarified in the manuscript. As stated in our response to comment #2.1.2, the following sentence has been added to the introduction:

“…we also describe other health outcomes to provide a full overview of the effect of two-way messages in an African context.”

#2.2.2 Eligibility Criteria, Please describe how (and with what search terms) two-way text message interventions were included in your search strategy; often you cannot tell whether an SMS intervention is 1 or 2 way unless a careful review of the intervention/report is undertaken. This should also be mentioned as a limitation of your review, in that it’s likely 2-way interventions were missed and others might describe two-way interventions differently than the authors did for this manuscript. 

Response #2.2.2: We thank the reviewer for pinpointing this issue. As described in the introduction we defined two-way text messages as a text message the receiver could react or respond to, with the use of a mobile phone, in any way. We have now also included this definition in our methodology section our “Eligibility Criteria”, which now states, 

“Trials comparing two-way text message interventions with standard care and/or one-way text messages were included. We defined two-way text messages as a text message the receiver could react or respond to, with the use of a mobile phone, in any way.”

First, as can be seen from our full search strings in the S1 file, our search strategy was not only focused on “two-way messages” but on text messaging in general, e.g. they also contained search terms that were broad terms such as “SMS” or “mobile phone”. Second, in the title and abstract screening we only excluded trials that obviously stated interventions consisting of one-way messages according to our definition. All other text message trials, where the type of intervention was uncertain, were read in full text to clarify the type of intervention. Based on the reviewer’s comments we have now clarified this and the “Information sources and search strategy” section now states:

“The search string included search terms such as “randomized controlled trial”, “Africa”, “SMS”, “mobile phone”, “two-way text messages” and “two-way SMS” (S1 File).”

AND 

“In the title-abstract screening, trials were only excluded if the title and abstract obviously stated that interventions consisted of one-way messages as according to our definition.”

#2.2.3 Page 5 Line 122, were all trials assessed for bias by two people - so 100% double coding for risk of bias? This is not clear from the text. 

Response #2.2.3: All trials were assessed for risk of bias independently by two authors. We have added “all trials” to the sentence to clarify this, 

“Two authors (ESØ, LSL) independently assessed all trials that were eligible for meta-analysis (trials reporting the outcomes “appointment attendance” and/or “medicine adherence”) for risk of bias using the Cochrane Risk of Bias Tool [15]. If consensus could not be reached, and arbiter (DSL) made the final decision.”

#2.3 Results: 2.3.1: Page 12 Descriptive Analyses, i think this paragraph would benefit from a revised subtitle and explanation of what is being presented in this section. Initially it seemed like it would be trials not included in other analyses, but this is not the case. Please clarify and revise.

#Response 2.3.1: We thank the reviewer for this suggestion and agree that it needs revision. We have revised the heading and it now states, 

“Descriptive analyses of trials not included in meta-analysis”

Further, we have added the following sentence to the descriptive analyses section,

“Eight trials reported other outcomes or reported results on medicine adherence and/or appointment attendance in a format that did not allow us to include the data in meta-analyses. Therefore, these outcomes were only analyzed descriptively.”

#2.4 Discussion: 2.4.1: Page 17 Line 351, I do not think that it is accurate to state that people who are deemed “illiterate” in some contexts would not benefit from text messages and interventions; we have found that people can be literate in some information contexts while “illiterate” in others - and that even people who are deemed “illiterate” in classic reading/writing literacy can have a high capability of understanding text messages. 

Response #2.4.1: We thank the reviewer for this important comment. We have revised the discussion so it now states:

“The illiteracy rate among adults in Sub-Saharan Africa was 35% in 2019, and such individuals may not benefit from the interventions to the same extent as literates [51]. Nevertheless, it is not given that people deemed illiterate in a classic reading and writing context cannot benefit from text message interventions as they may still be capable of understanding text messages.”

#2.4.2 Page 20 Line 410-411, I would suggest including a reference to the mERA checklist (BMJ 2016;352:i1174 http://dx.doi.org/10.1136/bmj.i1174) - this checklist contributes to standardization in reporting/comparisons between interventions, and ensuring that essential aspects of mHealth interventions are attended to. 

Response #2.4.2: We thank the reviewer for this suggestion. We have included the reference in our manuscript. The discussion now states:

“We recommend “The mHealth evidence reporting and assessment checklist” as a reporting guide for future trials, to support replication and improve the quality of mobile phone health intervention trials [55].” 

#2.5 Overall more description and discussion of two-way text messaging is needed throughout the paper.

Response #2.5: We thank the reviewer for this comment. We have revised the manuscript thoroughly according to the reviewers’ constructive comments and two-way messaging is both described and discussed more in-depth in the revised manuscript.

Reviewer 2

Comment

#A. This is a timely review of an important tool in the mobile health toolbox: two-way texting interventions for improved health outcomes. The paper is well written and well organized, as is expected from this highly qualified author group. It is also critical that these reviews, although not novel, are completed routinely as the evidence is growing and the technology changing rapidly. 

However, the included studies are few and the outcomes limited, reducing the potential impact of this review on public health practice. There are many reviews of texting interventions – as the authors note and cite – including many focused on HIV, as is the majority of this one. Could more health topics be included (malnutrition, diabetes, vaccinations, etc)? 

Response #A: We thank the reviewer very much for the supportive comment. We did not exclude any health outcomes in the paper but included all two-way text message trials set in Africa. Yet, we only conducted meta-analyses on trials that had the outcomes “medicine adherence” and “appointment attendance” (dichotomously measured), and trials that did not have such outcomes were assessed descriptively. The trials eligible for meta-analyses concerned “HIV”, “malaria” and “hypertension”. Effects of the text messages on these health topics were assessed in subgroup analyses according to our protocol. Health topics such as “diabetes” or “reproductive health knowledge” were analyzed descriptively (table 1).

We agree that the number of two-way text message trials set in Africa are limited, which reduces the potential impact of this review on public health practice. We have revised the discussion, so it now states:

“We included all types of health outcomes in this review and conducted meta-analyses on the outcomes “appointment attendance” and “medicine adherence” whilst other outcomes were described descriptively. The majority of included trials reported results on either medicine adherence or appointment attendance, yet this still only entailed that 10 trials were included our meta-analyses as the other were reported in a format that did not allow for inclusion. Further, eight out of ten trials included in the meta-analyses concerned “HIV”. This limits the generalizability of our findings beyond the field of HIV and reduce the potential impact of this review on public health practice.”

#B. Could the study assessment criteria extend beyond a more simple assessment of high or low bias to a more comprehensive and nuanced review of the study rigor? Furthermore, the call for more better evidence has been made for over a decade. The authors know and could give more specific suggestions on what is needed. For example, the authors could strengthen the paper by making more detailed suggestions for what could create an improved evidence base, potentially including focal areas in the WHO guide, “Monitoring and evaluating digital health interventions: A practical guide to conducting research and assessment” (https://www.who.int/reproductivehealth/publications/mhealth/digital-health-interventions/en/ ). Whether referring to that guide or other evaluation criteria, the authors could make a larger contribution to the digital health community if the review would focus more on the weaknesses of the included studies (or the excluded studies), moving past a more generic call for “more high quality trials.” That would exponentially increase the impact of this paper.

Response #B: We thank the reviewer for these constructive comments. We agree that the recommendation of better evidence is outdated and have now deleted it from our recommendations and have revised the discussion in line with the reviewer’s comments:

“Further, we only focused on intervention effects in relation to two health outcomes of interest and we did not focus on other aspects of mHealth interventions, such as the participants perspectives. As suggested by the WHO guide “monitoring and evaluating digital health interventions” and the model for assessment of telemedical applications (MAST), an evaluation of not only the clinical effectiveness but also other dimensions, which the technology affects, may provide a more nuanced understanding of how technological interventions truly work [51, 52].”

The conclusion now states:

“Two-way text messages seemingly improve medicine adherence but has an uncertain effect on appointment attendance and other healthcare outcomes compared to standard care. We recommend taking this into account when planning future digital health strategies in Africa. To better estimate the true impact of two-way text message interventions, we recommend that future trials and reviews also consider other aspects of technological interventions, such as feasibility, usability and sustainability as recommended by the WHO.”

#1. The words used for searching seem too exclusive. Other than, “appointment attendance” and “medicine adherence”, what other words were used? Strings like “visit attendance,” “treatment adherence,” and “linkage to care” might have also brought in trials. Were these key words also considered? Texting, cell phones or digital health may have brought up other studies. 

Response #1: We thank the reviewer for this comment. In our review, we included all health outcomes, hence, our search strings did not have any limitations in relation to appointment attendance and medicine adherence, which can be seen in the S1 file. We have clarified this further under “eligibility criteria”. It now states: 

“Trials reporting all health care outcomes were included as long as it met all other eligibility criteria.”

#2. Lines 49-51 and 51-53: break into 2 sentences for clarity.

Response #2: We thank the reviewer for this direct point. We have changed the sentence. It now states: 

“For the past 30 years, information and communication technologies have transformed the world [1]. In recent years mobile phone access has expanded immensely across Africa and other low- and middle-income countries (LMICs). According to the World Health Organization (WHO), more people have access to a mobile phone than to clean running water in sub-Saharan Africa [2]. In 2017, approximately 8 out of 10 people in sub-Saharan Africa owned a mobile phone with an increasing amount of these being smartphones [3].”

#3. Many sentences in the intro would benefit from a grammatical review.

Response #3: We thank the reviewer for this point. We have now gone through the manuscript and corrected grammatical errors.

#4. Line 85: males and females separate or also together?

Response #4:We thank the reviewer for pinpointing. We have now clarified it in the manuscript. It now states:

“We included randomized trials of male and female healthcare clients, partners, or guardians for healthcare clients, e.g., parents for child patients.”

#5. Line 150: what about for non dichotomous outcomes, i.e. % of on-time visits/year, for example?

Response #5: We thank the reviewer for this comment. The one trial that reported continuous outcomes on medicine adherence and appointment attendance was analyzed descriptively. We have clarified this in the manuscript. The methodology section now states,

“Trials that reported dichotomous outcome data in a format that did not allow inclusion in meta-analysis or trials that only reported continuous outcomes were analyzed descriptively”

#6. 154: what is I2 ? What is it or spell out.

Response #6: We thank the reviewer for this point. We have now made a reference to the methods behind the I2 statistic:

 “Heterogeneity were assessed using I2 [15]”

#7. Line 185: This does not make sense. How can infants be the target group for a mHealth innovation? Does this mean the guardian or parent of an infant?

Response #7: We thank the reviewer for this question. We have rephrased the sentence. It now states, 

“The primary target group for the text-message interventions were guardians of infant patients [19, 20, 24, 25, 36, 37, 39, 40, 46, 47] where the mother was the receiver of the text message intervention…”

#8. Line 213: This specific RCT was testing whether post-operative follow-up by two-way text messages was as safe as in-person visits for follow-up to identify and report adverse events. The intent was to use two-way messaging as a form of telehealth – comparing adverse events between the intervention arm as compared to standard care – trying to reduce workload of nurses. Two-way texting actually identified /more/ adverse events (a positive for quality care) as the men were provided with reassurance on wound care and encouraged to return to care if they had a problem. Two-way texting was not non-inferior (the texting actually improved reporting), and the lack of significant difference between the arms is a positive outcome for potential workload reduction and quality care. The original sentence starting on line 213, “Another trial investigated the effect of two-way text message intervention on circumcision harms in Zimbabwe and did not find any differences compared with standard care (risk difference (RD): 1.04% (p=0.32) [20],” is incorrect and misleading. Texting was not related to harms but on identifying potential harms. It should be replaced with something like, “Another trial investigated the effect of two-way text messages (intervention) as compared to routine in-person reviews (standard care) for post-operative follow-up among male circumcision patients in Zimbabwe and did not find any significant difference in adverse events between arms (risk difference (RD): 1.04% (p=0.32) [20]. However, although this study outcome [20] was not among those studies included in the primary outcomes of the overall meta-analysis, misunderstanding these study outcomes does give pause as to the veracity of the other findings reported in the paper. I am not going to review the content of each included paper, but the authors should be confident that the content of the meta-analysis and the study summaries are correct. 

Response #8: We thank the reviewer for pinpointing this error and apologize for not being clear in our conclusions of this trial. We have revised the paragraph as suggested,

“One trial investigated the effect of two-way text messages (intervention) compared to routine in-person visits (standard care) for post-operative follow-up among male circumcision patients Zimbabwe and did not find any significant difference in adverse events between groups (risk difference (RD): 1.04% (p=0.32)) [22].”

#9. Lines 225-227, what is the outcome of the study? The sentence tells only what the study investigated.

Response #9: We thank the reviewer for this point. We now have described the outcome more clearly,

“Further, one cluster randomized trial investigated the effect of two-way text messages that promoted contraception use and compared these to a control group receiving a sham text message (i.e. text messages with nutrition-focused-content) on the incidence of unintended pregnancies among sex workers in Kenya. They found no difference in the incidence of unintended pregnancy over 12 months of follow-up in the intervention group compared to the control group (hazard ratio (HR): 0.98; 95% CI: 0.7 to 1.4) [35].”

#10. Line 289: so all HIV studies were compared to the one hypertension study? What would a significant finding tell the reader anyway if almost all trials were for HIV?

Response #10: We thank the reviewer for this valid point. We conducted the subgroup analyses according to our protocol where we stratified the analysis across various clinical areas. The majority of trials included in our meta-analysis were HIV trials, which limits the generalizability of our findings. We have clarified this in our results and discussion sections. 

Results:

“When stratifying the primary analyses into different clinical areas, 8 out of ten trials concerned HIV care [19, 21, 24, 26, 28, 30, 31, 45] whilst the remaining two trails concerned hypertension [23] and malaria [21].”

AND

Discussion:

“Further, eight out of ten trials included in the meta-analyses concerned “HIV”. This limits the generalizability of our findings beyond the field of HIV and reduce the potential impact of this review on public health practice.”

#11. Line 298. What is meant by the effect of the call back versus text back option mean? Is this comparing whether the clients sent back an SMS or called back to confirm attendance? To confirm attendance intent? Please clarify.

12. Line 300: please clarify again what is meant by call back vs. SMS. This is critical to readers’ understanding of what these results may mean for future interventions. Does this mean giving clients a chance to call back was more effective for medicine adherence? 

Response #11+12: We thank the reviewer for these comments. It is clear to us that our classification of two-way text messages into “call back” and “text back” options causes confusion, as also pointed out by reviewer #1. The description now states, 

“Subgroup analyses were performed comparing overall low risk of bias trials with high risk of bias trials, clinical areas and types of interventions, e.g. if the receiver could interact with a health care professional (HCP) through a phone call or text message”

#13. Lines 334-338 lack clarity, “compared to standard care, two-way text messages slightly improved diabetes control [32], reproductive health knowledge [31, 39] post-partum contraceptive use [30], early breastfeeding [23] though not adverse events [22], unintended pregnancy [33] or HIV medicine adherence [42]. I think that citation [22] here should be [20]. First, this makes me request that you review all your citations within the paper. Second, if [22] is meant to be [20], please rephrase to, ….”early breastfeeding [23] and adverse event ascertainment [20] although not unintended pregnancy [33] or HIV medicine adherence [42].” However, isn’t this metareview about HIV medicine adherence? Why wasn’t [42] included above?

Response #13: We thank the reviewer for pinpointing this error. We have gone through all the references and made sure that they are correct. It is correct that reference 42 (now 41) included the relevant outcome. However, the trial results were reported in a format that did not allow inclusion in our meta-analysis. This is also stated under “Meta-Analysis: Primary analyses”: 

“One trial was excluded from the meta-analysis as it only reported continuous data on “medicine adherence” [32] and three trials did not report results in a format that allowed inclusion in meta-analysis [27, 41, 44].”

#14. Line 359: the restricted outcomes to only 2 (appointment attendance or medicine adherence) is also a large limitation. What other outcomes are the intended impact of two-way texting interventions? Testing uptake? Linkage to care? Self-monitoring? Smoking or alcohol harms reduction? Intimate partner violence? Improved nutrition or malnutrition identification?

Response #14: We thank the reviewer for this point. The issue was also raised by reviewer #1 (comment 2.1.2). Please see our response to reviewer #1’s comment.

#15. Line 365: text back options are not well explained. What does this mean? I thought that the previous lines (298-302) noted the impact of the call back? This is unclear, muddling a potential impact of this type of review on actual intervention development.

Response #15: We thank the reviewer for this comment, which is well in line with the reviewer’s previous comments #11 and #12. We have now rephrased our description of these messages as described above. 

#16. Line 402: Spell out WHO the first time

Response #16: This has now been done.

---

## [Decision Letter · Decision Letter 1]

28 Mar 2022

Two-way text message interventions and healthcare outcome in Africa: Systematic review of randomized trials with meta-analyses on appointment attendance and medicine adherence

PONE-D-21-10098R1

Dear Dr. Linde,

We’re pleased to inform you that your manuscript has been judged scientifically suitable for publication and will be formally accepted for publication once it meets all outstanding technical requirements.

Kind regards,

Michelle L. Munro-Kramer, PhD, CNM, FNP-BC

Academic Editor

PLOS ONE

Additional Editor Comments (optional):

Thank you for addressing all previous comments. Congratulations on the acceptance of this manuscript.

Reviewers' comments:

Reviewer's Responses to Questions

**Comments to the Author**

1. If the authors have adequately addressed your comments raised in a previous round of review and you feel that this manuscript is now acceptable for publication, you may indicate that here to bypass the “Comments to the Author” section, enter your conflict of interest statement in the “Confidential to Editor” section, and submit your "Accept" recommendation.

Reviewer #2: All comments have been addressed

2. Is the manuscript technically sound, and do the data support the conclusions?

Reviewer #2: Yes

3. Has the statistical analysis been performed appropriately and rigorously? 

Reviewer #2: Yes

4. Have the authors made all data underlying the findings in their manuscript fully available?

Reviewer #2: Yes

5. Is the manuscript presented in an intelligible fashion and written in standard English?

Reviewer #2: Yes

6. Review Comments to the Author

Reviewer #2: The paper has been well revised. I do not have any additional comments to the authors. Now, meeting character counts.

7. PLOS authors have the option to publish the peer review history of their article (what does this mean?). If published, this will include your full peer review and any attached files.

Reviewer #2: No